# Complete Removal of the Lesion as a Guidance in the Management of Patients with Breast Ductal Carcinoma In Situ

**DOI:** 10.3390/cancers13040868

**Published:** 2021-02-18

**Authors:** Luca Nicosia, Giuseppe di Giulio, Anna Carla Bozzini, Marianna Fanizza, Francesco Ballati, Anna Rotili, Matteo Lazzeroni, Antuono Latronico, Francesca Abbate, Giuseppe Renne, Francesca Addante, Marco Lucioni, Enrico Cassano, Mauro Giuseppe Mastropasqua

**Affiliations:** 1Department of Breast Radiology, IEO European Institute of Oncology IRCCS, 20141 Milan, Italy; luca.nicosia@ieo.it (L.N.); anna.bozzini@ieo.it (A.C.B.); anna.rotili@ieo.it (A.R.); antuono.latronico@ieo.it (A.L.); francesca.abbate@ieo.it (F.A.); enrico.cassano@ieo.it (E.C.); 2Department of Breast Radiology, Fondazione IRCCS—Policlinico San Matteo, 27100 Pavia, Italy; g.digiulio@smatteo.pv.it (G.d.G.); m.fanizza@smatteo.pv.it (M.F.); f.ballati@smatteo.pv.it (F.B.); 3Division of Cancer Prevention and Genetics, IEO European Institute of Oncology IRCCS, 20141 Milan, Italy; matteo.lazzeroni@ieo.it; 4Division of Pathology and Laboratory Medicine, IEO European Institute of Oncology IRCCS, 20141 Milan, Italy; giuseppe.renne@ieo.it; 5Department of Emergency and Organ Transplantation, Section of Anatomic Pathology, School of Medicine, University “Aldo Moro”, 70124 Bari, Italy; f.addante2@studenti.uniba.it; 6Department of Molecular Medicine, Anatomic Pathology, University of Pavia, Fondazione IRCCS Policlinico San Matteo, 27100 Pavia, Italy; m.lucioni@smatteo.pv.it

**Keywords:** ductal carcinoma in situ (DCIS), invasive breast carcinoma, underestimation, upgrade rate, vacuum-assisted breast biopsy (VABB), breast microcalcifications, active surveillance

## Abstract

**Simple Summary:**

A diagnosis of ductal carcinoma in situ, made on biopsy, is often followed by surgery or radiotherapy because of the risk of an upgrading disease upon subsequent surgical specimens, finding invasive carcinoma. In order to select which patients can be spared overtreatments and alternatively followed with active surveillance, we retrospectively reviewed 2173 vacuum assisted breast biopsies. Our goal was to demonstrate if complete removal of the lesion by biopsy, documented by mammograms, can be a valid criterion to select the patients that can be spared further treatments. The results of our study demonstrate a significant lower upgrading rate of disease when the lesion is completely removed. Thus, performing a mammogram to document the absence of residual lesion following vacuum-assisted breast biopsy (VABB) allows us to reduce overtreatments and to select which patients can be followed with an active surveillance, sparing unjustified public health costs.

**Abstract:**

*Background*: Considering highly selected patients with ductal carcinoma in situ (DCIS), active surveillance is a valid alternative to surgery. Our study aimed to show the reliability of post-biopsy complete lesion removal, documented by mammogram, as additional criterion to select these patients. *Methods*: A total of 2173 vacuum-assisted breast biopsies (VABBs) documented as DCIS were reviewed. Surgery was performed in all cases. We retrospectively collected the reports of post-VABB complete lesion removal and the histological results of the biopsy and surgery. We calculated the rate of upgrade of DCIS identified on VABB upon excision for patients with post-biopsy complete lesion removal and for those showing residual lesion. *Results*: We observed 2173 cases of DCIS: 408 classified as low-grade, 1262 as intermediate-grade, and 503 as high-grade. The overall upgrading rate to invasive carcinoma was 15.2% (330/2173). The upgrade rate was 8.2% in patients showing mammographically documented complete removal of the lesion and 19% in patients without complete removal. *Conclusion*: The absence of mammographically documented residual lesion following VABB was found to be associated with a lower upgrading rate of DCIS to invasive carcinoma on surgical excision and should be considered when deciding the proper management DCIS diagnosis.

## 1. Introduction

Breast cancer is the most common cancer diagnosed in the female population, accounting for approximately 15.2–30% of all new cancer cases among women.

Ductal carcinoma in situ (DCIS) of the breast represents a heterogeneous group of neoplastic lesions confined to the breast ducts and lobules, without showing invasive features nor metastatic potential [1].

About 25% of all breast cancer cases are ductal carcinoma in situ, and thus their diagnostic and therapeutic management represents an important health challenge with fundamental public health implications.

DCIS is usually diagnosed by imaging because it is often clinically occult. Its incidence has rapidly increased from 1980 considering the dramatic improvement in diagnosis and screening imaging tools. Mammography (Figure 1) plays a central role, since it is the cornerstone of breast cancer screening and diagnosis [2].

Nowadays, approximately 98% of patients with DCIS undergo surgery, often associated with radiotherapy [3]. However, it is now clear that most of them rarely progress spontaneously to invasive cancer, and indeed the mortality rate is as low as 4% [4]. The risk of progression seems to be related the grade of the disease, with high-grade tumor being associated with a worse prognosis [5,6]. Moreover, according to certain studies, a higher aggressiveness is due to multifocality as well as to aberrant branching and lobularization, defined as neoductgenesis [7,8]. Thus, the identification of these patterns at imaging and histology could help in distinguishing intrinsic aggressiveness and tailoring the therapy accordingly.

Therefore, we can assume that aggressive treatment of DCIS, especially in patients with additional pathologies, can be considered a form of overtreatment. Nevertheless, surgery and long follow-up periods are comparable in terms of public health costs [9].

Four prospective international study protocols (LORIS, COMET, LORD, and LORETTA) are currently in place to evaluate non-invasive treatment strategies for DCIS [10,11,12,13,14]. The main purposes of the abovementioned studies consist in examining the effectiveness and safety of active surveillance compared with surgical-based treatment approaches for low-risk DCIS patients [10] (Table 1).

The effectiveness of active surveillance can be improved by reducing the rate of upgrade—presurgical biopsy-proven DCIS may be upgraded to invasive carcinoma on submitted surgical specimens.

However, data regarding DCIS diagnostic underestimation rate are quite controversial—according to an important meta-analysis performed by Brennan et al., up to 26% of patients with biopsy-proven DCIS revealed a synchronous invasive carcinoma on surgical specimens [15].

The primary purpose of our observational multicenter retrospective study was to determine the rate of upgrade of DCIS identified on vacuum-assisted breast biopsy (VABB) upon excision and the possible relationship with the post-VABB complete removal of the lesion.

In order to do this, the residual tumor rate found on surgical specimen was compared with imaging of mammogram performed post-VABB but before subsequent surgery.

## 2. Materials and Methods

We reviewed all cases of breast biopsies with DCIS diagnosed on VABB at our Departments of Pathology from 1 January 2000 to 31 December 2018, and subsequent surgical excision performed in 2 medical centers (IEO, European Institute of Oncology, Milan, Italy, and “San Matteo” Hospital, Pavia, Italy).

Since VABB provides a better diagnostic performance than core needle biopsy [16], we selected patients submitted to this procedure using a 10G needle.

During the considered period of this study, we used a subcategorization of DCIS according to the so-called DIN (ductal intraepithelial neoplasia) system, as previously published [17]. Briefly, DIN1C corresponds to low-grade DCIS, DIN2 to intermediate-grade, and DIN3 to high-grade, according to nuclear morphologic features of the neoplastic cells [18,19].

Patients younger than 40 years of age, those with concomitant invasive carcinoma or past personal history of breast cancer, and those showing DCIS with comedonecrosis were excluded from the study.

All these data were retrospectively collected.

By using mammogram before surgery, we recorded the absence or the presence of post-VABB residual lesion and we compared the outcomes of these 2 groups of patients.

The upgrade rate of DCIS to invasive carcinoma following surgical excision was always recorded.

### Statistics

Fisher’s exact text was performed to evaluate the difference between the proportions of the upgrade rate to invasive cancer on surgical excision with and without macrospical residual lesion after biopsy.

All analyses were performed with the statistical software SAS 9.4 (SAS Institute, Cary, NC, USA). Categorical data are reported as counts and percentages.

*p*-values less than 0.05 were considered as statistically significant.

## 3. Results

A total number of 2173 vacuum-assisted breast biopsies were performed under stereotactic guidance showing DCIS: 408 cases were low-grade (DIN1C), 1262 cases were intermediate-grade (DIN2), and 503 cases were high-grade (DIN3). The mean age of the patients was 62 years (range 32–84 years). The overall mean diameter of the lesions was 20 mm. Table 2 summarizes clinicopathologic characteristics of the patients.

Taken as a whole, 15.2% (330/2173) of DCISs were upgraded to invasive cancer on surgical excision.

We observed post-VABB the complete removal of the lesion in 785 out of 2173 (36.1%) patients. By considering this subgroup, we reported 65 cases of invasive carcinoma on surgical specimen, and thus 8.3% (65/785) of DCIS were upgraded to invasive cancer. The mean diameter of the lesion removed with the biopsy was 20 mm.

These data led to the first observation—patients showing complete removal of the lesion experienced a significantly lower upgrade rate compared to those showing mammographically detectable residual tumor after VABB (*p*-value < 0.05).

Data considering the three diagnostic categories (DIN1C, DIN2, and DIN3) are summarized in Table 3.

### 3.1. DCIS Subcategories

#### 3.1.1. DIN1C (Low-Grade DCIS)

We observed that 408 patients received the diagnosis of DIN1C (low-grade DCIS, Figure 2)—9.6% (39/408) of them were upgraded to invasive cancer. The overall mean diameter of the DIN1C lesions was 22 mm.

We reported, post-VABB, complete removal of the lesion in 159 out of 408 patients with DIN1C diagnosis. Among them, we reported nine cases of invasive carcinoma on surgical specimen, and thus 5.7% (9/159) of low-grade DCIS cases with no residual lesion were upgraded to invasive cancer.

Patients with diagnosis of low-grade DCIS showing complete removal of the lesion experienced a significantly lower upgrade rate when compared to those showing mammographically detectable residual tumor after VABB (*p*-value < 0.05).

#### 3.1.2. DIN2 (Intermediate-Grade DCIS)

We observed that 1262 patients received the diagnosis of DIN2 (intermediate-grade DCIS, Figure 3)—15.1% (191/1262) of them were upgraded to invasive cancer. The overall mean diameter of the DIN2 lesions was 20 mm.

We reported, post-VABB, complete removal of the lesion in 420 out of 1262 patients with DIN2 diagnosis. Among them, we reported 33 cases of invasive carcinoma on surgical specimen, and thus 7.8% (33/420) of intermediate-grade DCIS cases with no residual lesion were upgraded to invasive cancer.

Patients with diagnosis of intermediate-grade DCIS showing complete removal of the lesion experienced a significantly lower upgrade rate compared to those showing mammographically detectable residual tumor after VABB (*p*-value < 0.05).

#### 3.1.3. DIN3 (High-Grade DCIS) 

We observed that 503 patients received the diagnosis of DIN3 (high-grade DCIS, Figure 4)—19.9% (100/503) of them were upgraded to invasive cancer. The overall mean diameter of the DIN3 lesions was 25 mm.

We reported, post-VABB, complete removal of the lesion in 206 out of 503 patients with DIN3 diagnosis. Among them, we reported 23 cases of invasive carcinoma on surgical specimen, and thus 11.2% (23/206) of high-grade DCIS cases with no residual lesion were upgraded to invasive cancer.

Patients with diagnosis of high-grade DCIS showing complete removal of the lesion experienced a significantly lower upgrade rate compared to those showing mammographically detectable residual tumor after VABB (*p*-value < 0.05).

## 4. Discussion

Considering that most of DCIS will never progress to invasive breast cancer during a patient’s lifetime, surgical therapy and radiotherapy of DCIS, especially in patients with comorbidities, can be considered a form of overtreatment, without taking into account unjustified health and social care costs.

Surveillance, epidemiology, and end results (SEER) data show that the 20-year breast cancer-specific mortality rate in patients with DCIS is as low as 3.3% [4,9]. On the other hand, according to a significant meta-analysis, 25.9% (18.6–37.2%) of presurgical cases diagnosed as DCIS were upgraded to invasive carcinoma upon excision [15].

In this study, we tried to minimize the risk of diagnostic underestimation by applying strict inclusion criteria. In particular, in our series, all cases were biopsied by VABB with at least a 10G needle; patients younger than 40 or patients with previous history of breast cancer were excluded.

Our overall upgrading rate of 15.2% was in line with other previous studies [21,22,23] that have reported upgrading rates in the range 11–25% (Table 4).

Furthermore, in order to reduce the diagnostic underestimation rate as much as possible, we took into account additional parameters such as the post-biopsy complete removal of the lesion [24], information not reported in other studies, and the diameter of the lesion, information evaluated only in the LORETTA trial (diameter < 25 mm) [14].

The results of our study show that if the lesion is completely removed during biopsy, the overall diagnostic underestimation rate is significantly lower. Indeed, DCIS patients showing complete removal of the lesion experienced a significantly lower upgrade rate to invasive cancer compared to those showing mammographically detectable residual tumor after VABB (8.2% vs. 19%, respectively).

However, although this difference is significant, the clinical relevance is debatable, since patients showing residual lesion on mammogram still have a chance (81%) to not upgrade.

Therefore, we strongly believe that this last parameter should be considered as a possible selection criterion to offer DCIS patients an active surveillance program.

As far as we know, this study has one of the largest number of biopsies considered in a single retrospective study, but the main limitation is its retrospective nature. Moreover, involved patients do not perfectly match inclusion criteria of LORIS, LORETTA, COMET, and LORD protocols.

## 5. Conclusions

The absence of mammographically documented residual lesion following VABB is associated with a lower upgrading rate of DCIS to invasive carcinoma on surgical specimens and should be taken into account when deciding the proper management of patients with ductal carcinoma in situ diagnosis.

## Figures and Tables

**Figure 1 cancers-13-00868-f001:**
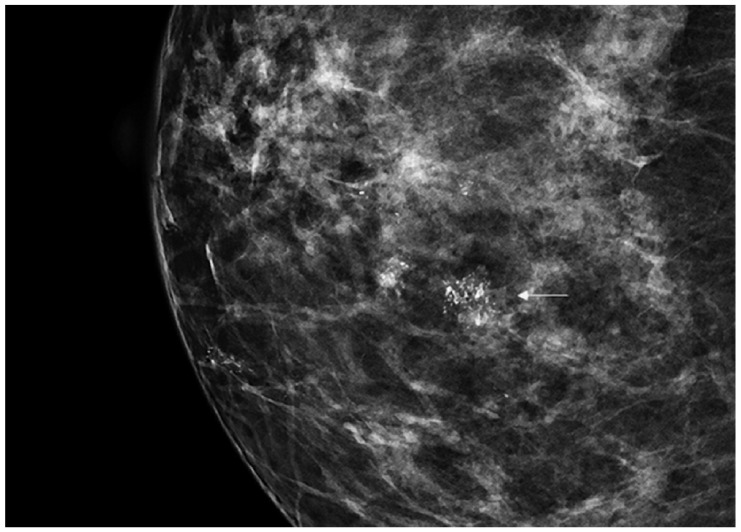
Spot magnification mammogram with a small cluster of pleomorphic microcalcification (arrow) suspicious for ductal carcinoma in situ (DCIS)/ductal intraepithelial neoplasia (DIN).

**Figure 2 cancers-13-00868-f002:**
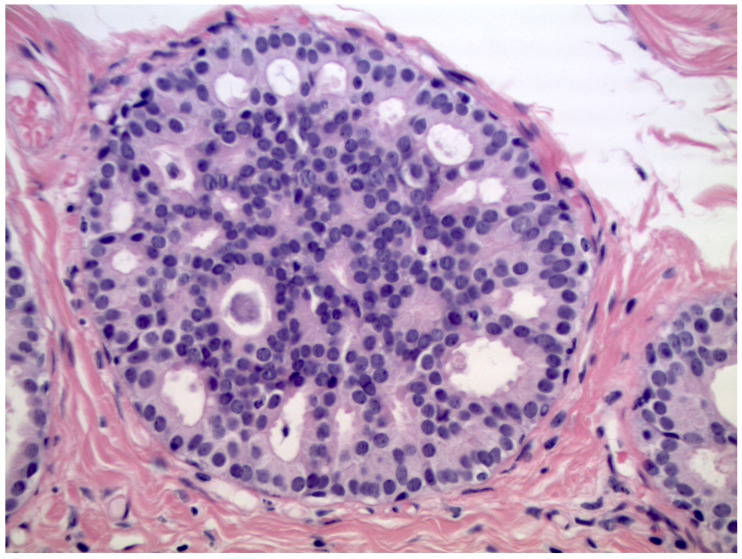
Cribriform ductal carcinoma in situ of low nuclear grade (Hematoxylin & Eosin, 400×).

**Figure 3 cancers-13-00868-f003:**
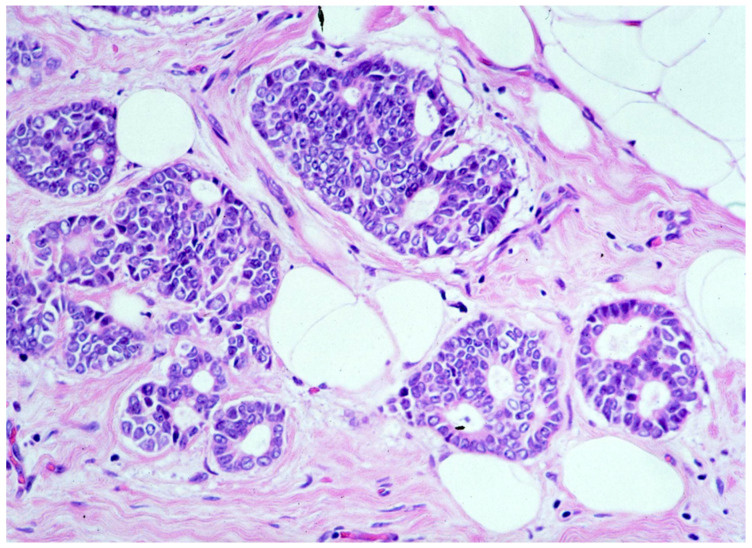
Ductal carcinoma in situ of intermediate nuclear grade (DIN2).

**Figure 4 cancers-13-00868-f004:**
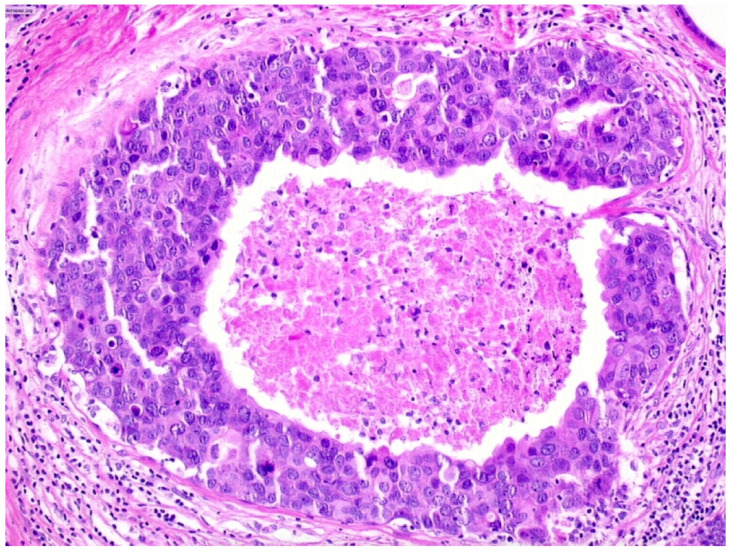
Ductal carcinoma in situ of high nuclear grade with central necrosis (DIN3).

**Table 1 cancers-13-00868-t001:** Main aspects of the four prospective international study protocols (LORIS, COMET, LORD, and LORETTA).

Study	LORIS [11]	COMET [12]	LORD [13]	LORETTA [14]
Country	UK	USA	EU	Japan
Year of activation	2014	2017	2017	2017
Accrual target (number of patients)	932	1200	1240	340
Minimum age at diagnosis (years)	48	40	45	40
Comedonecrosis	Excluded	Allowed	Excluded	Excluded
Hormone receptor status	Any	HR-positive only	Any	HR-positive only
Size of the lesion	Any	Any	Any	<2.5 cm
Type of guide for biopsy	Stereotactic (vacuum-assisted)	Stereotactic (vacuum-assisted)	Stereotactic (vacuum-assisted)	Stereotactic and ultrasound (vacuum-assisted)
Endocrine therapy	Optional	Optional	Not allowed	Mandatory

**Table 2 cancers-13-00868-t002:** Patient characteristics.

Clincico-Pathologic Features	DIN1C	DIN2	DIN3	Overall
Patient number	408	1262	503	2173
Age at VABB, mean (years)	50 (40–82)	54 (43–87)	49 (44–85)	54 (40–87)
Mean diameter of the lesion (mm)	22 (5–75)	20 (7–60)	25 (4–80)	20 (5–80)
BIRADS (Breast Imaging-Reporting and Data System) [20]				
BIRADS 3	10 (2.4%)	2 (0.2%)	0 (0%)	12 (0.5%)
BIRADS 4a	308 (75.6%)	952 (75.5%)	10 (1.9%)	1270 (58.4%)
BIRADS 4b	60 (14.7%)	248 (19.6%)	102 (20.2%)	410 (18.9%)
BIRADS 4c	27 (6.6%)	50 (3.9%)	345 (68.7%)	422 (19.4%)
BIRADS 5	3 (0.7%)	10 (0.8%)	46 (9.2%)	59 (2.7%)
Absence of residual disease post-VABB	159 (39%)	420 (33.3%)	206 (41%)	785 (36.1%)
Family history	230 (56.3%)	754 (59.7%)	330 (65.6%)	1314 (60.5%)

**Table 3 cancers-13-00868-t003:** Diagnostic underestimation rate comparison between cases with and cases without residual lesion post-biopsy.

Residual Disease Status of Diagnostic Categories					Comments	*p*-Value for Testing Differences between the Two Proportions (Absence and Presence of Residual Disease)
Absence of Residual Disease Post-Biopsy (DIN1C)					Percentage of upgrading rate from DCIS to invasive disease = 5.7%	*p* < 0.05
	Final Surgical Evaluation					
VABB Result	Negative	DIN1C	IN	Total		
DIN1C	19	131	9	159		
Presence of Residual Disease Post-Biopsy (DIN1C)					Percentage of upgrading rate from DCIS to invasive disease = 12%	
	Final Surgical Evaluation					
VABB Result	Negative	DIN1C	IN	Total		
DIN1C	43	176	30	249		
Absence of Residual Disease Post-Biopsy (DIN2)					Percentage of upgrading rate from DCIS to invasive disease = 7.8%	*p* < 0.05
	Final Surgical Evaluation					
VABB Result	Negative	DIN2	IN	Total		
DIN2	49	338	33	420		
Presence of residual disease post biopsy (DIN2)					Percentage of upgrading rate from DCIS to invasive disease = 18.7%	
	Final surgical evaluation					
VABB Result	Negative	DIN2	IN	Total		
DIN2	34	650	158	842		
Absence of Residual Disease Post-Biopsy (DIN3)					Percentage of upgrading rate from DCIS to invasive disease = 11.2%	*p* < 0. 05
	Final Surgical Evaluation					
VABB Result	Negative	DIN3	IN	Total		
DIN3	14	169	23	206		
Presence of Residual Disease Post-Biopsy (DIN3)					Percentage of upgrading rate from DCIS to invasive disease = 25.9%	
	Final Surgical Evaluation					
VABB Result	Negative	DIN3	IN	Total		
DIN3	31	189	77	297		
Absence of Residual Disease Post-Biopsy (Overall)					Percentage of upgrading rate from DCIS to invasive disease = 8.2%	*p* < 0.05
	Final Surgical Evaluation					
VABB Result	Negative	DIN	IN	Total		
Overall	72	638	65	785		
Presence of Residual Disease Post-Biopsy (Overall)					Percentage of upgrading rate from DCIS to invasive disease = 19%	
	Final Surgical Evaluation					
VABB Result	Negative	DIN	IN	Total		
Overall	108	1015	265	1388		

VABB: vacuum-assisted breast biopsy; IN: invasive neoplasia; DIN: ductal epithelial neoplasia.

**Table 4 cancers-13-00868-t004:** Upgrading rates to invasive carcinoma of breast biopsies in different studies.

References	No. Patients	Years	Biopsy Type	Upstaging Rate to Invasive Cancer
Brennan et al. (meta-analysis of 52 studies) [15]	7350	1996–2011	Variable	26% overall 21% non-high-grade 32% high-grade
Soumian et al. [21]	225	2001–2010	VABB	18% overall 10% low-grade 23% high-grade
Pilewskie et al. [22]	296	2009–2012	Variable	8% low-grade 22% intermediate-grade
Grimm et al. [23]	307	2008–2015	VABB	17% overall 7% low-grade 7% intermediate-grade 23% high-grade
Current study	2173	2000–2018	VABB	15.2% overall 9.6% low-grade 15.1% intermediate-grade 19.3% high-grade
Current study (post-biopsy removal of the lesion)	2173	2000–2018	VABB	8.2% overall 5.6% low-grade 7.8% intermediate-grade 11.1% high-grade

## Data Availability

Data sharing not applicable.

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
