# Peer review of "Complete Removal of the Lesion as a Guidance in the Management of Patients with Breast Ductal Carcinoma In Situ"

_cancers, 2021, doi:10.3390/cancers13040868_

Round 1
Reviewer 1 Report
My comments are attached in a seperate file.

Author Response
We thank the Reviewer for the relevant and useful suggestions that have improved our work.
Please, attached you'll find our point-to-point response to the reviewer's comments, which are also copied-and-pasted here for Your convenience.
This is a well written manuscript reporting the frequency of upgrading to IBC after a biopsy diagnosis of pure DCIS. Although many previous manuscripts have been published about this topic, they focus on the presence of residual disease on mammograms, with adds novel information to current literature
We thank the Reviewer for the encouraging comments.
Point 1: Overall: they reported a significant difference in upgrading between patients with versus without residual disease (8.2 versus 19%). However, although this difference is significant, the clinical relevance is questionable since patients without residual disease on mammagram still have 8.2% risk of upgrading and patients with residual disease have a chance of 81% that there is no upgrading. This point should be mentioned in the discussion.
Response 1: The Reviewer’s comment is very pertinent. We have revised our discussion taking into consideration the abovementioned observation.
Indeed, we added the comparison between the upgrading rate in patients with and without residual lesion in the discussion section (lines 206-208).
Point 2: Abstact: The upgrade was lower (8.2%) in patients with mammographically complete removal compared to patients without complete removal (19%). This percentage of 19% should be added, since it provides additional information next to the overall % of 15.2.
Response 2: The reviewer’s suggestion has been accepted and we have modified the Abstract at lines 48-49.
Point 3: The authors state that treatment leads to unjustified costs, but is treatment indeed more expensive than follow-up during a long period of the patients life?
Response 3: The Reviewer’s comment is appropriate, and we have reconsidered our opinion modifying the text in the Introduction section (lines 79-81).
Point 4: Methods: I do not understand why they excluded patients with comedonecrosis.
Response 4: We have excluded comedonecrosis in order to be in line with most (3 out of 4) of the prospective international study protocols cited in Table 1.
Point 5: Table 3: The p-values are unclear, there are two p-values next to each item, what do they mean? In addition, the P-values in the text do not correspond with the P-values in the Table 3 (for example P<0.019 in line 158 of the text). Please check all these P-values and clarify
Response 5: We apology for the mistake. The p values of Table 3 and in the text have been accordingly changed and matched.
Point 6: Discussion, line 205: 8.2% versus 19% instead of 8.2% versus 15.2%.
Response 6: We again apology for the mistake. The value has been corrected.

Reviewer 2 Report
This manuscript address the biopsy-related 88 diagnostic underestimations in a large number of patients. They retrospectively reviewed 2173 vacuum as-31 sisted breast biopsies and found that the upgrading rate of DCIS to invasive carcinoma is low (5.6%, 9/159) when the lesion is completely removed. The results may provide a guideance on patient management with breast cancer (ductal carcinoma In Situ) to avoid overtreatment.
Overall, the manuscript is technically sounds and easy to understand. However, there are issues need to be fixed. First, the title reads like a review article instead of research one. It is too broad, and I should summarize the overall content instead of its extension. Second, it will help the readers better unserdstand the cancer clarifications, typical mammogram and pathological images of DCIS should be presented for each category(DIN1C, DIN2, DIN3). Third, the statistical analysis methodology is oversimplified, details should be given in the methodology section.
Belows are some minor issues to be addressed:
Line 57-65: It read like most of the information came from a single reference, which may not be accurate and ideal.
Line 67: "mammography which plays a central role" The "which" reads oddly. Please consider two sentences instead of a complex one.
Line 68: this sub-sentence is not a reason why "mammography which plays a central role".
Line 78-79: "patients with comorbidities" is not clear. Please explain it.
Line 83-85: What is the "main proposal"?
Line 95: What is the multicenter?
Line 96: What is the difference between "rate of upgrade" and "biopsy-related diagnostic underestimation"?
Line 98: "The secondary objective of the study is to prove a correlation". There are no methods and results on the "correlation" that fulfill this goal in the following sections.
Line 107: two centers: medical centers or two tenters from specimens?
Line 109: Why 12G needle
Line 115-117: please give a reason why these exclusions are necessary.
Line 122: what does the upstage rate mean? upgrade?
Line 125: Please give details of the "proportions test". How the test performed?
Line 133: BIRADS.Please spell out the abbreviation, and give the proper references.
Line 154: 5.7% (9/159): it was 5.6% in table 3. Please be consistent.
Line 158: p-value < 0.019. This number is not consistent with the number in table 3.
Line 175: 11.2% is not consistent with the 11.1% in table 3.
Line 207: "enrol"?
Author Response
We thank a lot the Reviewer for the valuable opinions and comments.
We have taken into account all the suggestions modifying our main text.
Here, please, you will find our point-to-point responses.
This manuscript address the biopsy-related 88 diagnostic underestimations in a large number of patients. They retrospectively reviewed 2173 vacuum as-31 sisted breast biopsies and found that the upgrading rate of DCIS to invasive carcinoma is low (5.6%, 9/159) when the lesion is completely removed. The results may provide a guideance on patient management with breast cancer (ductal carcinoma In Situ) to avoid overtreatment.
Overall, the manuscript is technically sounds and easy to understand. However, there are issues need to be fixed. First, the title reads like a review article instead of research one. It is too broad, and I should summarize the overall content instead of its extension. Second, it will help the readers better unserdstand the cancer clarifications, typical mammogram and pathological images of DCIS should be presented for each category (DIN1C, DIN2, DIN3). Third, the statistical analysis methodology is oversimplified, details should be given in the methodology section.
First. We thank the Reviewer for the suggestion. We have changed the title as follow:
Complete removal of the lesion as a guidance in the management of patients with breast ductal carcinoma in situ.
Second. We have added radiological and pathological pictures as suggested.
Third. We have expanded the Methods section, adding details of the adopted statistical methodology.
Belows are some minor issues to be addressed
Line 57-65: It read like most of the information came from a single reference, which may not be accurate and Ideal
We thank the Reviewer for the valuable observation. Indeed, we agree and we have revised the references, deleting two of them (i.e. reff 2,3 of the original draft).
Line 67: "mammography which plays a central role" The "which" reads oddly. Please consider two sentences instead of a complex one.
The Reviewer’s point of view is fitting, and we changed the text creating two separate sentences.
Line 68: this sub-sentence is not a reason why "mammography which plays a central role".
We have revised the sub-sentence, and we have modified the text “…since it is the cornerstone of breast cancer screening and diagnosis”
Line 78-79: "patients with comorbidities" is not clear. Please explain it.
We referred to patients with other additional pathologies that could jeopardize the post-surgical care. We have changed the term comorbidities in favor of “additional pathologies”.
Line 83-85: What is the "main proposal”?
We agree with this comment and we have rephrased the text in order to clarify the issue.
Line 95: What is the multicenter?
The procedures were performed in two centers: European Institute of Oncology (IEO), Milan, Italy and Polyclinic “San Matteo”, Pavia, Italy.
We have accordingly changed the text.
Line 96: What is the difference between "rate of upgrade" and "biopsy-related diagnostic underestimation"?
Effectively, we agree that using different terms may be confounding. In order to clarify the issue, we have deleted “biopsy-related diagnostic underestimation” and replaced it with “rate of upgrade”.
Line 98: "The secondary objective of the study is to prove a correlation". There are no methods and results on the "correlation" that fulfill this goal in the following sections.
The Reviewer’s comment is very appropriate. We apologize for the mistake. It was one of our primary goal of the study and it has not been deleted in the final draft.
We have removed that sentence, because we did not find any correlation and modified the previous one.
Line 107: two centers: medical centers or two tenters from specimens?
We realized the unclarity of the sentence.
Actually, we meant two medical centers (European Institute of Oncology, Milan, Italy and Polyclinic San Matteo, Pavia, Italy).
We have accordingly changed the text.
Line 109: Why 12G needle
We apologize for the typing mistake. All procedures were performed using a 10G needle.
In our institutes most of VABBs are performed using a 10G needle and, sometimes, an 8G needle. For this study we excluded the 8G biopsies in order to get a uniform cohort of patients.
We changed the text accordingly.
Line 115-117: please give a reason why these exclusions are necessary.
We tried to be in line with the criteria of the four clinical trials we cited in the paper.
We excluded patients younger than 40 years as well as in the COMET trial; for the same reason we excluded patients with concurrent invasive carcinoma. Past personal history of breast cancer is an exclusion criterion for LORIS trial. Comedonecrosis is allowed only in COMET trial.
Line 122: what does the upstage rate mean? upgrade?
We apologize for the typo. As the Reviewer correctly interpreted, we meant upgrade and we have corrected the text.
Line 125: Please give details of the "proportions test". How the test performed?
With the proportion text we meant Fisher exact test. The Fisher exact text was performed to evaluate the difference between the proportions of the upgrade rate to invasive cancer on surgical excision with and without macroscopical residual lesion after biopsy.
All analyses were performed with the statistical software SAS 9.4 (SAS Institute, Cary, NC). Categorical data are reported as counts and percentages.
P-values less than 0.05 were considered as statistically significant.
Line 133: BIRADS.Please spell out the abbreviation, and give the proper references.
We apologize for the inappropriate use of abbreviation. BIRADS means Breast Imaging-Reporting and Data System and we have accordingly modified the text, adding the proper reference.
Line 154: 5.7% (9/159): it was 5.6% in table 3. Please be consistent.
We apologize for the typo. The correct value is 5.7%. We changed the mistaken value in the table.
Line 158: p-value < 0.019. This number is not consistent with the number in table 3.
We apologize for the mistake. As we reviewed all the statistical analysis, we correct the p-value in both table and text.
Line 175: 11.2% is not consistent with the 11.1% in table 3.
We again apologize for the typo. The correct value is 11.2%. We changed the mistaken value in the table.
Line 207: "enrol"?
We agree with the Reviewer about the ambiguity of the term and we have rephrased the sentence as follow: “…possible selection criterion to offer DCIS patients an active surveillance programme”.
